# Identification of Myofascial Trigger Point Using the Combination of Texture Analysis in B-Mode Ultrasound with Machine Learning Classifiers

**DOI:** 10.3390/s23249873

**Published:** 2023-12-16

**Authors:** Fatemeh Shomal Zadeh, Ryan G. L. Koh, Banu Dilek, Kei Masani, Dinesh Kumbhare

**Affiliations:** 1Institute of Biomedical Engineering, University of Toronto, Toronto, ON M5S 3G9, Canada; fatemeh.shomalzadeh@mail.utoronto.ca (F.S.Z.); k.masani@utoronto.ca (K.M.); 2KITE Research Institute, Toronto Rehabilitation Institute, University Health Network, Toronto, ON M5G 2A2, Canada; ryan.koh@mail.utoronto.ca; 3Department of Physical Medicine and Rehabilitation, Dokuz Eylul University, Izmir 35340, Turkey; banu.dilek@deu.edu.tr

**Keywords:** myofascial trigger point, texture features, machine learning, ultrasound

## Abstract

Myofascial pain syndrome is a chronic pain disorder characterized by myofascial trigger points (MTrPs). Quantitative ultrasound (US) techniques can be used to discriminate MTrPs from healthy muscle. In this study, 90 B-mode US images of upper trapezius muscles were collected from 63 participants (left and/or right side(s)). Four texture feature approaches (individually and a combination of them) were employed that focused on identifying spots, and edges were used to explore the discrimination between the three groups: active MTrPs (*n* = 30), latent MTrPs (*n* = 30), and healthy muscle (*n* = 30). Machine learning (ML) and one-way analysis of variance were used to investigate the discrimination ability of the different approaches. Statistically significant results were seen in almost all examined features for each texture feature approach, but, in contrast, ML techniques struggled to produce robust discrimination. The ML techniques showed that two texture features (i.e., correlation and mean) within the combination of texture features were most important in classifying the three groups. This discrepancy between traditional statistical analysis and ML techniques prompts the need for further investigation of texture-based approaches in US for the discrimination of MTrPs.

## 1. Introduction

Chronic pain (e.g., myofascial pain syndrome (MPS)) affects nearly one hundred million adults in the United States with an annual cost between USD 560 to 635 billion [1]. MPS is one of the most prevalent musculoskeletal pain disorders that occur in every age group and has been associated with primary pain conditions, including osteoarthritis, disc syndrome, tendinitis, migraines, and spinal dysfunction [2]. Myofascial trigger points can be used to characterize MPS. These can be split into two types: active MTrPs (A-MTrP), which are spontaneously painful nodules, and latent MTrPs (L-MTrP), which are nodules that are only painful when palpated.

MTrPs have been classically defined as a “hyperirritable spot” in skeletal muscle that is associated with a hypersensitive palpable nodule in a taut band [3]. The diagnostic criteria for MPS involve physical screening, but studies have shown that the manual detection of MTrPs is unreliable [4]. Quantitative techniques can help improve the detection of MTrPs.

Ultrasound (US) is an attractive modality for this problem as it has been used to identify MTrPs [4,5,6]. It is a non-invasive way to assess muscles, tendons, and ligaments [7,8,9] and is relatively low cost. Doppler and elastography US have been used to visualize and distinguish MTrPs from normal tissue [8,10,11,12]. Unfortunately, not all clinical US machines are equipped with elastography capabilities, and these approaches require comprehensive training to use and interpret. Brightness mode (B-mode) US, on the other hand, is readily available in most clinics and hospitals and would be the preferred option for diagnosing and screening musculoskeletal disorders if possible.

However, B-mode US has high variability in echo intensity depending on the operator, model, and more. Thus, texture features have been used to mitigate this issue and have been widely used to discriminate variables in B-mode US images. Texture features play a vital role in radiomics, providing information such as muscle fiber orientation, normal anatomy, and the extent of adipose, fibrous, and other connective tissues within muscle [10]. Previous studies have suggested that the muscle fibers within the MTrPs in the affected zone and the muscle fibers in the surrounding regions have different orientations in comparison with normal skeletal muscle [10].

Although texture feature analysis of US images has been explored to distinguish MTrPs in affected muscle from normal tissue [8,11,12], there is currently no “gold standard” to detect the presence of MTrPs within B-mode US images. Previous research has used various methods of analyzing texture to tackle this problem, such as using entropy characteristics [11], gray-level co-occurrence matrices (GLCM), blob analysis, local binary pattern (LBP), and statistical analysis [12]. A comprehensive review paper on texture analysis or classification categorized these techniques into four main categories [13]:**Transform-Based:** Transform-based techniques employ a set of predefined filters or kernels to extract texture information from an image. Common filters include Gabor filters and LBP [14,15]. These filters highlight certain frequency components or local variations in pixel values, making them suitable for tasks where patterns are characterized by specific spatial frequencies or orientations.**Structural:** Structural techniques focus on describing the spatial arrangement and relationships between different elements in an image. They often involve identifying and characterizing specific patterns or structures within the texture (e.g., GLCM). These methods are valuable for capturing details related to texture regularity, directionality, or organization.**Statistical:** Statistical methods involve the analysis of various statistical properties of pixel intensities within an image or a region of interest (ROI). Common statistical features include entropy, contrast, correlation, homogeneity, energy, mean, and variance. These metrics quantify the distribution and variation of pixel values, providing insights into the texture’s overall properties, such as roughness, homogeneity, or randomness.**Model-Based:** Model-based methods involve fitting mathematical or statistical models to patterns in an image. These models can be simple, such as a parametric distribution (i.e., Gaussian distribution or Markov random fields), or more complex, such as deep learning models like convolutional neural networks. Model-based approaches are versatile and can capture intricate texture patterns, making them increasingly popular for texture analysis.

Of these categories, we focused on features that may better describe spots, edges, and patterns. This is because a variety of studies describe MTrPs as “knots” in the muscle. A wide variety of studies describe the MTrPs as a hyperechoic band, hypoechoic elliptical region, or simply a different echo architecture than the surrounding muscle tissue in clinical examination (e.g., US screening) [16,17].

For many clinicians and investigators, the finding of one or more MTrPs is required to assure the diagnosis of MPS. However, there remains a lack of optimal methods for characterizing these muscle structures, and achieving an objective characterization of MTrPs has the potential to enhance their localization and diagnosis. This can facilitate the development of clinical measures [15]. One of the leading challenges in the classification of B-mode US images is that they may vary in scale, view, or intensity. For these reasons, various approaches attempt to address these challenges.

Gabor filters are a feature that can be used to detect direction and are often used to reveal lines and edges in an image [18]. They can also be used to determine the structure and visual content contained within an image [13]. Previously, Gabor filters have been used to enhance fiber orientation and detect edges in US images [19,20]. LBP is another approach that can be used to characterize skeletal muscle composition in patients with MPS compared with normal healthy participants [8,21]. Most of these approaches have been used for image processing and statistical analysis, but classification may benefit greatly from the incorporation of machine learning (ML).

ML approaches may enhance classification as they are able to autonomously learn patterns and relationships from data [22,23,24]. ML is focused on making predictions as accurate as possible, while traditional statistical models are aimed at inferring relationships between variables [24]. ML offers advantages in terms of flexibility and scalability when contrasted with conventional statistical methods, allowing its utilization across various tasks like diagnosing, classifying, and predicting survival. Nevertheless, it is crucial to assess and compare the accuracy of muscle characterization through traditional statistical methods and ML within the context of clinical screening [25]. Supervised ML algorithms (e.g., neural networks (NNs), decision trees (DTs), etc.) can generalize from training data to make accurate predictions or classifications on new, unseen data. Their adaptability allows them to handle diverse domains and tasks, making them invaluable tools for tasks ranging from image recognition to medical diagnosis, enhancing efficiency and precision in decision-making processes [22].

Thus, this study delves into the utilization of various texture feature approaches and ML techniques to classify and characterize MTrPs in US images. We investigate different texture feature approaches (i.e., LBP, Gabor, SEGL method, and their combination with texture features) extracted from US images to classify MTrPs. We further employ various ML techniques as well as traditional statistical analysis to explore the effectiveness of the extracted features from the US images to characterize and classify the muscle between A-MTrPs, L-MTrPs, and healthy muscle.

## 2. Materials and Methods

### 2.1. Participants

Participants (*n* = 63) were recruited from the musculoskeletal/pain specialty outpatient clinic at the Toronto Rehabilitation Institute. The upper trapezius muscle of all participants was examined. All participants underwent a physical examination by a trained clinician on our team (BD), who determined the presence or absence of MTrPs (i.e., A-MTrPs and L-MTrPs) in the upper trapezius muscle according to the standard clinical criteria defined by Travell and Simons [3] and through visual confirmation on B-mode US. Participants who demonstrated no symptoms or history related to neuromuscular disease, based on diagnostic criteria, were included in this study. Each participant’s muscle(s) (right and/or left) was labeled as A-MTrPs (*n* = 30), L-MTrPs (*n* = 30), or healthy control (*n* = 30) (Table 1).

All subjects gave their informed consent for inclusion before they participated in the study and their upper trapezius muscles were included in our study. The study was conducted in accordance with the Declaration of Helsinki, and the protocol was approved by the Institutional Review Board of the University Health Network (UHN) (protocol code 15-9488).

### 2.2. Ultrasound Acquisition Protocol and Pre-Processing

US videos were acquired using a US system (SonixTouch Q+, Ultrasonix Medical Corporation, Richmond, BC, Canada) with a linear ultrasonic transducer of 6–15 MHz and a depth set to 2.5 cm. The acquisition settings including time gain compensation, depth, and sector size were held constant across all participants. Acquisition was performed by an experienced sonographer with the participant sitting upright in a chair with their arms relaxed on their sides and forearms resting on their thighs. The transducer was placed on the skin in the center of the trapezius muscle (i.e., the midpoint of the muscle belly between the C7 spinous process and the acromioclavicular joint) with enough gel to cover the entire surface (Figure 1). A ten-second video (sampling frequency: 30 frames/second) of the trapezius muscle from each side per participant was recorded by moving the transducer towards the acromioclavicular joint (parallel to the orientation of the muscle fibers) at approximately 1 cm/s, generating 300 images per participant for analysis (Figure 1). While recording the video, the researcher manipulated the transducer’s position to reduce artifacts and mitigate muscle distortion caused by the transducer, such as applying downward pressure. From each video, 4 unique frames/images were manually selected out of 300 B-mode images. These selected images captured various sections of the muscle (i.e., lateral to medial) and were used to validate the presence or absence of MTrPs evident in the video (Figure 2A). Images from each side of a participant (e.g., left and/or right trapezius) were treated as independent sites (Table 1).

ROIs of the muscle (i.e., the region between the upper trapezius muscle’s superior and inferior fascia) were manually extracted from the acquired images via visual localization. These muscle ROIs were further analyzed using the following texture features (Figure 3).

### 2.3. Texture Feature Analyses

**I. Local Binary Patterns.** LBP, a rotationally invariant feature, is one of the most popular texture feature analysis operators [26]. It can evaluate the local spatial patterns and contrast of grayscale images. This technique calculates eigenvalues for the different patterns in an image, such as edges and corners within a neighborhood. LBP was calculated for every B-mode image using the following equation below (Equation (1)).
(1)$$LBP_{P,\;\;R}=\sum_{p=0}^{P-1}s\left(g_{p}-g_{c}\right)2^{p}\;,\;s\left(x\right)=\left\{\begin{array}{c} 1,\;x\geq 0\\ 0,x<0, \end{array}\right.\;$$
where P is the number of pixels within the neighborhood and within a circle radius of R = 1, $g_{p}$ represents the *p*th neighboring pixel, $g_{c}$ represents the center pixel, and $s\left(x\right)$ is the obtained binary code at position (*x*) neighbors.

In our study, a 3 by 3 neighborhood was used, and its central pixel intensity was compared with its surrounding eight neighbor pixels [27]. If the neighboring pixel intensity was below the pixel intensity of the central pixel, then it was labeled 0; otherwise, it was assigned the value 1. This resultant binary matrix was then multiplied by a fixed weight matrix, which was then summed replacing the central pixel (i.e., the LBP measure). This produced one of 256 (2^8^) possible patterns.

LBP was calculated across the entire ROI, and the outer border of the ROI (i.e., did not have eight neighbors) was replaced with the next closest pixel values (Figure 2C).

**II. Gabor Feature.** Gabor filtering was introduced by Daugman and used in pattern analysis applications [28,29,30]. The Gabor filter-based features are directly extracted from the gray-level images (i.e., B-mode images) and compute a measure of “energy” in a window around each pixel in each response image. In the spatial domain, a two-dimensional Gabor filter is a Gaussian kernel function modulated by a complex sinusoidal plane wave, defined as (Equation (2)) [31]:(2)$$G\left(x,y\right)=exp\left\{\frac{-1}{2}\;\left(\frac{x^{\prime }{}^{2}}{\sigma_{1}^{2}}+\frac{x^{\prime }{}^{2}}{\sigma_{2}^{2}}\;\right)\right\}cos\left(2\pi ƒx^{\prime }+\varphi \right),$$
$$x^{\prime }=xsin\theta +ycos\theta ,\;\;y^{\prime }=-xcos\theta +ysin\theta ,$$
where *ƒ* is the spatial frequency of the wave at angle *θ* with the x-axis, $\sigma_{1}$ and $\sigma_{2}$ are the standard deviations of the 2-D Gaussian envelope, and φ is the phase.

Gabor features were calculated using the Gabor feature extraction function created by Haghighat et al. [32] in MATLAB (2023a, The MathWorks, Natick, MA, USA). Forty Gabor filters were calculated at 5 frequency scales for eight orientations (i.e., *θ*: 0°, 45°, 90°, 135°, 180°, 225°, 270°, and 315°), producing 40 Gabor feature images for each B-mode US image (Figure 2B).

**III. SEGL Method.** SEGL stands for statistical, edge, GLCM, and LBP and was proposed by Fekri Ershad S. for textual analysis [33]. It is a feature extraction method that combines statistical, edge, GLCM, and LBP features. First, LBP is calculated from the input image. Then, GLCM is calculated on the resultant LBP image in which the edge feature is then calculated before calculating the statistical features.

GLCM was proposed by Haralick and Shanmugam [34]. GLCM provides information about how often a pixel with the intensity value i occurs in a specific spatial relationship to a pixel with the value j. In this study, GLCM was calculated along 8 directions (i.e., *θ*: 0°, 45°, 90°, 135°, 180°, 225°, 270°, and 315°) with an empirically determined distance (offset = one pixel).

Edge detection is the process of localizing pixel intensity transitions that have been used to extract information in the image via object recognition, target tracking, segmentation, etc. It is defined by a discontinuity in gray-level values or a boundary between two regions with relatively distinct gray-level values [35]. The Canny edge detection method was used, as previous literature has shown that the Sobel edge detection method cannot produce smooth and thin edges compared to the Canny method [36].

Finally, the seven statistical features, described below (Section 4), were calculated over the edge detected images. This resulted in 56 features (8 directions × 7 statistical features).

**IV. Statistical Feature.** Statistical features were used to measure the image variation. In our study, 7 statistical features of entropy, energy, mean, contrast, homogeneity, correlation, and variance were computed. A summary of these statistical features is provided below (Equations (3)–(9)).

Entropy: shows the degree of randomness of pixel intensities within an image (Equation (3)) [7,30,34].


(3)
$$Entropy:a_{1}=-\sum_{i,j=0}^{N-1}\mathrm{Ln}\left(p_{i,j}\right)\;p_{i,j},$$


Contrast: measures the local contrast of an image (Equation (4)).


(4)
$$Contrast:a_{2}=\sum_{i,j=0}^{N-1}P_{i,j}({i-j)}^{2},\;$$


Correlation: provides a correlation between two pixels in a pixel pair (Equation (5)).


(5)
$$Correlation:a_{3}=\sum_{i,j=0}^{N-1}P_{i,j}({i-\mu )(j-\mu )/\sigma }^{2}),$$


Homogeneity: measures the local homogeneity of a pixel pair (Equation (6)).


(6)
Homogeneity:a4=∑i,j=0N−1Pi,j1+i−j2


Energy: measures the number of repeated pairs (Equation (7)).


(7)
$$Energy:a_{5}=\sum_{i,j=0}^{N-1}{{(P}_{i,j})}^{2}\;$$


Mean (Equation (8)):


(8)
$$Mean:a_{6}=\sum_{i,j=0}^{N-1}{i\;(P}_{i,j})\;$$


Variance (Equation (9)):(9)$$Variance:a_{7}=\sum_{i,j=0}^{N-1}{P_{i,j}\;\left(i-\mu \right)}^{2}\;$$
where *P_i,j_* is the pixel value in position (*i*, *j*) in the output image, μ and σ are, respectively, the mean and standard deviation (variance) of all *P_i,j_* values in the output image, and N is the number of gray levels in the output image.

### 2.4. Classification Techniques, Training, and Evaluation

The features calculated from each approach (Table 2) were used to train a variety of ML models to discriminate muscle with MTrPs (A-MTrPs and L-MTrPs) from healthy muscle. ML models were implemented in Python using the Scikit Learn library. These ML models were logistic regression (LR) [37], decision tree (DT) [38], random forest (RF) [39], k-nearest neighbors (kNN) [40], naive Bayes (NB) [41], support vector machine (SVM) [42], and artificial neural networks (NNs) [43,44]. These models were used because they are common in the literature [45], have different strengths, and could easily be implemented. Each method used the libraries’ default parameters and other hyperparameters such as the number of neighbors in kNN, which was tuned using grid search (Table 3).

The NN was a single hidden-layer network (512) with a dropout layer (50% of nodes dropped). All activation functions were Rectified Linear Units. The output layer was a 3-node output with an activation function of SoftMax. The NN was trained for 250 epochs with an early stopping criterion of 7 epochs of no improvement in the validation loss. The learning rate was the default set by Keras and was decreased by a factor of 0.1 after 3 epochs of no improvement to a minimum learning rate = 0.00001.

Input to all classifiers were the features from each approach as seen in Table 2. A leave-one-site-out approach was used due to the low number of images to better evaluate performance. The remaining examples were used for training (i.e., LR, DT, RF, k-NN, NB, and SVM) with the exception of the NN approach, where they were split into 75% training and 25% validation sets. For example, a training set would consist of 356 images (89 sites × 4 US images), and a test set would consist of 4 images. In the case of the NN, the training and validation sets would consist of 268 and 88 images (67 sites × 4 US images; 22 sites × 4 US images), respectively. Performance was evaluated using classification accuracy, F1-score, sensitivity, specificity, positive predictive value (PPV), and negative predictive value (NPV), which were calculated via the function *statsOfMeasure* in MATLAB.

### 2.5. Ensemble Approaches, Feature Importance, and Statistical Analysis

The ML techniques were further investigated using an ensemble approach. The best-performing trained classifier for each technique (e.g., kNN, SVM, etc.) was selected based on the mean performance across all 4 feature approaches (i.e., B-mode LBP, Gabor, and SEGL), as shown with asterisks (*) in Table 3. These selected classifiers were then used to perform a majority vote for a classification task. This was implemented via the function *majorityvote* in MATLAB.

In addition, to determine which features were more important toward the classification task, we examined the classification performance of using a single statistical feature (e.g., entropy, mean, etc.) and removing a single feature (from the set of 7).

For the single statistical feature case, we took the features from all approaches (as seen in Table 2) and used only the statistical feature of interest. In cases where there were more than 7 features (i.e., Gabor and SEGL), the mean values were used (i.e., 40 entropy features converted into a single mean entropy feature for the Gabor approach). This resulted in a vector of 4 values (i.e., entropy feature from the 4 approaches).

For the removal of a single feature case, the same procedure was used except that the features that were not removed were used as inputs (i.e., vector of 24 values (6 statistical features × 4 approaches)). Statistical analysis was performed on each feature using a one-way analysis of variance (ANOVA) to compare the 3 groups: A-MTrPs, L-MTrPs, and healthy control.

## 3. Results

Table 4 shows the classification accuracy (%), F1-score, sensitivity, specificity, PPV, and NPV for the best parameter of each ML technique, with the bolded values showing the best performance for each parameter for each approach (B-mode, LBP, Gabor feature, and SEGL).

Figure 4 shows the confusion matrices of the ML techniques for each approach (B-mode, LBP, Gabor feature, and SEGL), each approach with the majority vote, a single statistical feature, and the removal of a single statistical feature. For each analysis, the ML classifier with the parameter that presented the best performance is shown in Table 5.

Table 5 shows the classification accuracy (%), F1-score, sensitivity, specificity, PPV, and NPV for the ensemble approach and the effects of using a single statistical feature and the removal of a single statistical feature. The highest performance can be seen with the “correlation” feature (accuracy = 53.33%, F1-score = 0.4861) and the removal of variance (accuracy = 51.67%, F1-score = 0.518).

Table 6 shows the results of the statistical analysis (mean and standard deviation) of all four approaches (B-mode, SEGL, Gabor, and LBP) between all 3 groups: A-MTrPs, L-MTrPs, and healthy controls with bolded values showing statistical significance. Statistical differences (*p* < 0.05) were seen for almost all features for all approaches except in B-mode (i.e., entropy, contrast, and energy) and Gabor (i.e., mean and correlation).

## 4. Discussion

Our study investigated the effectiveness of combining texture features derived from US images that focused on edges and spots for the purposes of discriminating muscles with MTrPs from healthy muscle.

Our findings indicate that a combined approach did not achieve a high level of accuracy in distinguishing between A-MTrPs, L-MTrPs, and healthy muscle. The combined approach showed slightly better performance (in majority votes) for the B-mode and SEGL method compared to the LBP and Gabor feature (49.44% and 49.44% vs. 47.22% and 48.89%, respectively). We hypothesized that structural and statistical approaches and a combination of them could better classify muscle with MTrPs from healthy muscle. However, the overall accuracies obtained from these combination approaches exhibited a similar range, ranging from 43.33% to 53.33%. These results are comparable to another study that compared texture features to a CNN approach [46]. Their F_1_-score ranged from 0.383 to 0.477 for their texture approaches (i.e., first-order statistical, LBP, and blob analysis) when classifying these three groups using an NN. This study shows better performance in the texture feature approach, which may be attributed to the combined ensemble approach and features that focus on structural information (i.e., spots and edges).

Additionally, when a simple ensemble approach using majority voting was used, almost no improvements were observed in the different approaches (i.e., SEGL classification accuracy: 48.05% to 49.44%, LBP classification accuracy: 48.89% to 47.22%, B-mode classification accuracy: 53.06% to 49.44%, and Gabor classification accuracy: 48.33 to 48.89%).

It is worth mentioning that, while the PPV and classification accuracy only showed an approximately 50% ability to distinguish MTrPs (i.e., A-MTrPs and L-MTrPs) from healthy muscle, the specificity and NPV results demonstrated almost 75%. This may be helpful in providing clinicians with more certainty in identifying the absence of MTrPs.

Statistical analysis showed no statistically significant differences in “correlation” and “mean” with respect to the Gabor feature approach (*p* = 0.0857 and *p* = 0.2338). This could be attributed to the fact that the Gabor feature measures the gray level of US images [47], and there were similar mean and standard deviations seen in the A-MTrP and L-MTrP groups as indicated in Table 5. These findings align with previous studies that have reported muscle with MTrPs to exhibit anisotropy [10].

While the statistical analysis revealed statistically significant differences in most features among the three groups, the ML techniques could not classify the three groups sufficiently. This may be due to the fact that the features are relatively overlapped among the three groups as seen in Table 5.

The result of our traditional statistical analysis agrees with the results seen in previous literature [8]. One study using LBP and blob analysis demonstrated statistically significant results between healthy individuals and patients with MPS (*p* < 0.001) [8]. Based on this, they suggested that a combination of texture features (i.e., LBP and blob area and count) can be used to describe differences between individuals with MPS and healthy individuals using a principal component analysis. However, this study grouped individuals with both A-MTrPs and L-MTrPs into the group of individuals with MPS. Koh et al. demonstrated better performance in classifying MTrPs (i.e., A-MTrP and L-MTrP grouped) from healthy muscle compared to the three-group case (i.e., A-MTrP, L-MTrP, and healthy muscle) [46]. These studies within the literature plus the results seen in this study suggest that MTrPs can be distinguished from healthy muscle but may not be sufficient for discrimination between the two types of MTrPs (i.e., A-MTrPs from L-MTrPs).

Notably, the ‘correlation’ and ‘mean’ features demonstrated better discriminatory ability than the other features, yielding accuracies of 53.33% and 52.5%, respectively. Unsurprisingly, when these features were removed, the accuracies decreased to the lowest values of 49.17% and 50.83%, respectively, suggesting that these features carry significant weight in the classification performance.

Overall, MTrPs have been identified and labeled as hypoechoic (dark grey) nodules in US images in previous literature [48,49]. However, recent research has proposed the identification of MTrPs as large hypoechoic contracture knots, which also exhibit smaller hyperechoic “speckles” within the hypoechoic contracture knot [50,51]. The presence of these “speckles” can affect the structural information of MTrPs within the muscle ROI and interfere with the characterization of muscle with MTrPs using texture feature analysis. For instance, entropy is capable of describing homogeneity and randomness in the observed patterns in US images, while LBP depicts the structural elements (spots, edges, etc.) of US backscatter. Consequently, the presence of different patterns within muscles affected by MTrPs may lead to variations in the values of calculated texture features within each group, thereby reducing the predictive power of the ML techniques.

Another aspect to consider is the relationship between the US image and the clinical scenario. The existing literature has proposed certain clinical criteria for MTrPs, but these criteria have not been clearly associated with specific US abnormalities. Currently, most researchers in this field concur that MTrPs are a physical entity that exhibits a spherical or elliptical shape, but this has not been thoroughly investigated [52]. Therefore, it is crucial to identify characteristics that can identify the MTrP in ultrasound, which can then be exploited for classification purposes.

In addition, defining the “border zone” that separates this region from the surrounding normal muscle is necessary, as previous literature has suggested that this border or transition zone may provide more valuable information than the lesion (i.e., hypoechoic contracture knot) itself [53]. Moreover, in cases where a patient experiences pain but does not present with MTrPs, it is uncertain if there is an ‘at-risk’ area that later transforms into a visually defined spherical/elliptical MTrPs.

To the best of our knowledge, this study represents the first attempt to investigate the combination of texture features focusing on information that represents the known representation of MTrP in US for discriminating muscle with MTrPs from healthy muscle. Previous studies have primarily relied on traditional statistical methods as opposed to ML approaches [12,54]. This study focused on a data-driven process relying less on user knowledge to achieve more precise predictions. This helps to avoid the mistake of using an inappropriate statistical model on the dataset, which could limit accuracy [24].

It is worth mentioning that the proposed approach of using a combination of texture features may be a potential tool in discriminating and characterizing the muscular structural information in various medical fields of activity. For example, a study used the features of entropy and energy in LBP images to quantitively assess the spastic biceps brachii muscle in post-stroke patients [55]. Additionally, a study used the angular second moment, contrast, and homogeneity features calculated over a GCLM feature in US images of the quadriceps to measure the muscle texture (pattern) under the effects of neuromuscular electrical stimulation to characterize individuals with lower back pain [56]. Thus, it is likely that the proposed approach could be used to interpret the uniformity of muscle patterns and abnormalities in other applications (e.g., rehabilitation).

### Limitations

One limitation of this study lies in the proper definition and localization of the region of MTrPs within the US images from the muscle for the analysis. While the literature agrees that MTrPs present as hypoechoic structures in US, it is uncertain what area around these regions constitutes the MTrP. Thus, the entire ROI of the muscle was used for analysis to ensure no information was missed, but this may not be an optimal approach.

Furthermore, while hypoechoic images are generally associated with hyperperfused areas and hyperechoic images with hypoperfused areas [57], it is important to acknowledge the possibility of image artifacts, such as anisotropy, in our patient population. Hypoechoic areas may also arise from acoustic shadowing behind calcifications, lymph nodes, and certain pathological conditions. However, in this study, the manual selection of images aimed to alleviate the presence of any artifacts.

## 5. Conclusions

In conclusion, this paper sheds light on the utilization of texture features and combining them in different approaches (i.e., statistical features with B-mode, Gabor, LBP, and SEGL method) for the classification of A-MTrPs, L-MTrPs, and healthy muscle. The focus was to capture structural information such as edges, spots, and other relevant features. In comparison to traditional statistical analysis methods (e.g., ANOVA), the employed ML classification techniques did not achieve high classification results, likely due to the significant overlapping observed among the statistical values between the groups (maximum reported accuracy of 53.33%). Nevertheless, our developed ML algorithms were mainly able to perform better when there were no MTrPs (e.g., identify the healthy muscles (true negative results)). The results, however, were still much higher than chance, suggesting that these groups may be distinguishable, but further investigation is required to improve either the features or technique for classification.

Therefore, this study highlights the need to explore the potential of extracting advanced texture features in combination with non-traditional statistical analysis for effectively identifying MTrPs from healthy muscle. Such endeavors can contribute to the development of more robust diagnostic criteria based on US image characteristics. The findings from these future studies hold promise for the development of improved mechanisms to aid in the accurate identification and diagnosis of MTrPs.

## Figures and Tables

**Figure 1 sensors-23-09873-f001:**
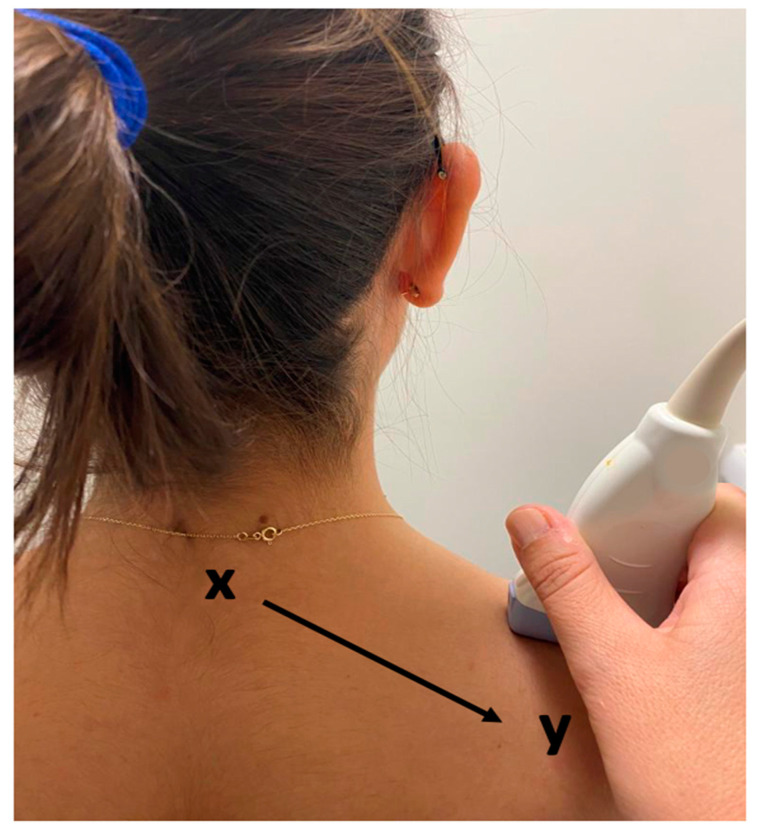
US transducer location from upper trapezius muscle (x = C7, y = acromion).

**Figure 2 sensors-23-09873-f002:**
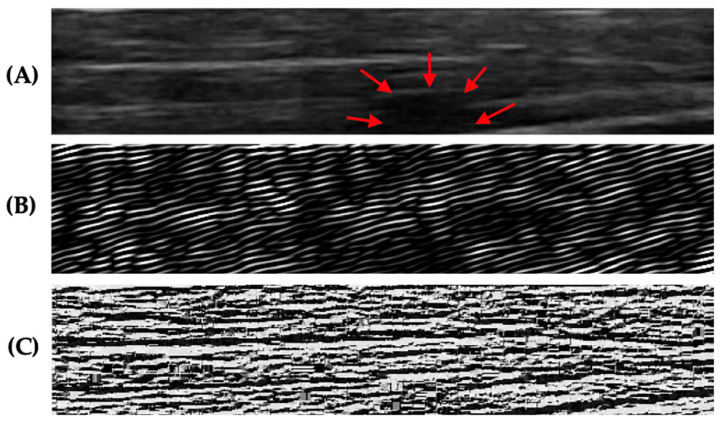
(**A**) An example of a B-mode US image from a participant with active MTrP. The red arrows show the MTrP, a hypoechoic region. (**B**) An example of a corresponding Gabor-filtered image (at *θ* = 0 degree) from the same participant. (**C**) An example of a corresponding LBP from the same participant.

**Figure 3 sensors-23-09873-f003:**
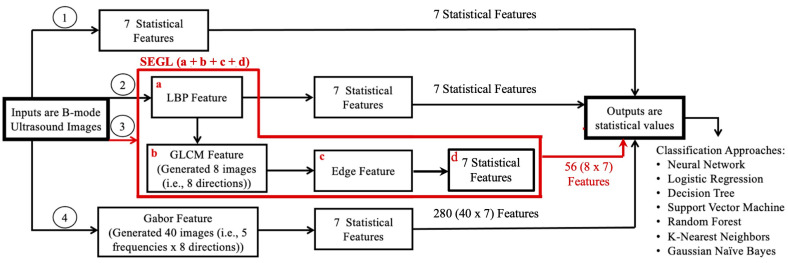
This chart shows a summary of the methods that were used for feature extraction. The red color connections represent the SEGL method, a combination of statistical, edge, and gray-level co-occurrence matrices (GLCM), and local binary pattern (LBP). Note: The numbers in each circle represent each approach.

**Figure 4 sensors-23-09873-f004:**
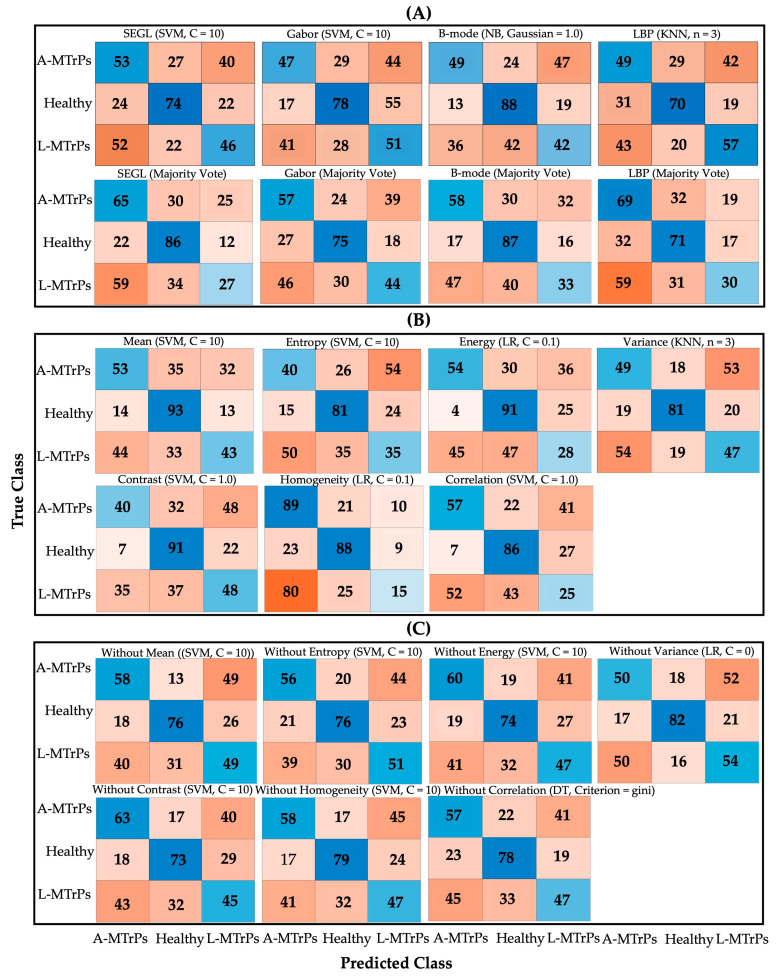
Confusion matrices of the ML algorithms with the best performance for (**A**) each approach (B-mode, LBP, Gabor feature, and SEGL) and each approach with the majority vote; (**B**) a single statistical feature; and (**C**) the removal of a single statistical feature for discriminating the three groups: A-MTrPs, L-MTrPs, and healthy controls.

**Table 1 sensors-23-09873-t001:** Number of participant’s muscles in each group.

Group	Number of Sites
A-MTrPs	30
L-MTrPs	30
Healthy Control	30

Note: Number of sites shows the number of each participant’s left and/or right muscle.

**Table 2 sensors-23-09873-t002:** The summary of approached texture features for each image (LBP, Gabor, SEGL, and LBP).

Approach	Number of Features
I. LBP	7
II. Gabor Feature	280 (40 × 7)
III. SEGL Method	56 (8 × 7)
IV. Statistical Features	7

**Table 3 sensors-23-09873-t003:** The following ML classifier techniques with their associated parameters were used. * Shows the best accuracy performance for each classifier technique.

Classifier Techniques	Hyperparameters
K-nearest neighbors (kNN) [39]	n_neighbors = 3, 5 *, 7
Decision tree (DT) [37]	Criterion = ‘gini’ *, ‘entropy’, ‘log_loss’
Random forest (RF) [38]	Criterion = ‘gini’ *, ‘entropy’, ‘log_loss’
Logistic regression (LR) [36]	C = 0.1, 1, 10 *
Naive bayes (NB) [40]	$\mathrm{Gaussian}\;(\mathrm{var\_smoothing}=1.0,\;10^{-5}$$,\;10^{-9}\;$*)
Support vector machine (SVM) [41]	C = 0.1, 1, 10 *
Artificial neural network (NN) [42,43]	

**Table 4 sensors-23-09873-t004:** Classification accuracy (%), F1-score, sensitivity, specificity, positive prediction values (PPVs), and negative prediction values (NPVs) for the best parameter of each ML technique for each approach (SEGL, LBP, B-mode, and Gabor).

Approach	ML Technique, Parameter	Accuracy (%)	F1-Score	Sensitivity	Specificity	PPV	NPV
SEGL Method	SVM, C = 10	**48.05**	0.4806	0.4806	0.7403	0.4814	0.7398
	LR, C = 10	46.39	0.4639	0.4639	0.7319	0.4644	0.7317
	DT, Criterion = ‘gini’	41.67	0.4168	0.4167	0.7083	0.4258	0.7056
	RF, Criterion = ‘log_loss’	45.56	0.4556	0.4556	0.7278	0.4703	0.7228
	KNN, N-neighbors = 5	43.33	0.4333	0.4333	0.7167	0.4315	0.7157
	NB, Gaussian, smoothing = 1.0	45.56	0.4556	0.4556	0.7278	0.4505	0.7081
	NN	44.44	0.4444	0.4444	0.7222	0.4467	0.7215
LBP	SVM, C = 10	44.17	0.4417	0.4417	0.7208	0.4447	0.7200
	LR, C = 1.0	45.56	0.4556	0.4556	0.7278	0.4629	0.7249
	DT, Criterion = ‘gini’	40.28	0.4028	0.4028	0.7014	0.3968	0.7026
	RF, Criterion = ‘log_loss’	45.28	0.4528	0.4528	0.7264	0.4555	0.7256
	KNN, N-neighbors = 3	**48.89**	0.4894	0.4889	0.7444	0.4879	0.7447
	$\mathrm{NB},\;\mathrm{Gaussian},\;\mathrm{smoothing}=10^{-5}$	40.00	0.4000	0.4000	0.7000	0.4138	0.6896
	NN	43.33	0.4333	0.4333	0.7167	0.4445	0.7132
B-mode	SVM, C = 0.1	52.22	0.5222	0.5222	0.7611	0.5278	0.7372
	LR, C = 1.0	45.83	0.4583	0.4583	0.7292	0.4710	0.7234
	DT, Criterion = ‘gini’	44.17	0.4417	0.4417	0.7208	0.4450	0.7196
	RF, Criterion = ‘gini’	49.72	0.4868	0.4972	0.7486	0.5088	0.7431
	KNN, N-neighbors = 5	50.83	0.5083	0.5083	0.7542	0.5108	0.7534
	NB, Gaussian, smoothing = 1.0	**53.06**	0.5306	0.5306	0.7653	0.5355	0.7460
	NN	46.94	0.4694	0.4694	0.7347	0.4858	0.7283
Gabor Filter	SVM, C = 10	**48.33**	0.4848	0.4889	0.7444	0.4945	0.7424
	LR, C = 10	45.00	0.4500	0.4500	0.7245	0.4515	0.7245
	DT, Criterion = ‘gini’	45.00	0.4500	0.4500	0.7250	0.4542	0.7237
	RF, Criterion = ‘log_loss’	46.67	0.4667	0.4667	0.7333	0.4777	0.7297
	KNN, N-neighbors = 5	47.22	0.4722	0.4722	0.7361	0.4757	0.7350
	$\mathrm{NB},\;\mathrm{Gaussian},\;\mathrm{smoothing}=10^{-5}$	43.61	0.4361	0.4361	0.7181	0.4341	0.7183
	NN	43.06	0.4306	0.4306	0.7153	0.4343	0.7141

Note: The bolded numbers represent the best performance for each approach.

**Table 5 sensors-23-09873-t005:** The classification accuracy (%), F1-score, sensitivity, specificity, PPV, and NPV for the ML techniques: for each approach (B-mode, LBP, Gabor feature, and SEGL) with the majority vote (highlighted in orange), a single statistical feature (highlighted in green), and the removal of a single statistical feature (highlighted in blue). (SVM: support vector machine, LR: logistic regression, KNN: K-nearest neighbors, DT: decision tree).

Approach/Feature	Accuracy (%)	F1-Score	Sensitivity	Specificity	PPV	NPV
SEGL Method (Majority Vote)	**49.44**	0.4731	0.4944	0.7472	0.5034	0.7384
LBP (Majority Vote)	47.22	0.4582	0.4722	0.7361	0.4703	0.7311
B-Mode (Majority Vote)	**49.44**	0.4786	0.4944	0.7472	0.5078	0.7397
Gabor Filter (Majority Vote)	48.89	0.4855	0.4889	0.7444	0.4922	0.7429
Entropy (SVM, C = 10)	43.33	0.4248	0.4333	0.7167	0.4472	0.7125
Energy (LR, C = 0.1)	48.06	0.4614	0.4806	0.7403	0.4993	0.7309
Contrast (SVM, C = 1)	49.72	0.4831	0.4972	0.7486	0.5082	0.7415
Correlation (SVM, C = 1)	**53.33**	0.4861	0.5333	0.7667	0.525	0.7485
Variance (KNN, K = 3)	49.17	0.405	0.4917	0.7458	0.4901	0.7462
Homogeneity (LR, C = 0.1)	46.67	0.4508	0.4667	0.7333	0.4882	0.7258
Mean (SVM, C = 10)	52.5	0.51	0.525	0.7625	0.5359	0.7551
Without Entropy (SVM, C = 10)	50.83	0.507	0.5083	0.7542	0.5107	0.7535
Without Energy (SVM, C = 10)	50.28	0.5014	0.5028	0.7514	0.5053	0.7507
Without Contrast (SVM, C = 10)	50.28	0.5014	0.5028	0.7514	0.5051	0.7507
Without Correlation (DT, Criterion = gini)	49.17	0.4868	0.4917	0.7458	0.4983	0.7435
Without Variance (LR, C = 10)	**51.67**	0.518	0.5167	0.7583	0.5135	0.759
Without Homogeneity (SVM, C = 10)	51.11	0.509	0.5111	0.7556	0.5149	0.7545
Without Mean (SVM, C = 10)	50.83	0.5088	0.5083	0.7542	0.5071	0.7544

Note: The bolded numbers represent the best performance for the ML algorithms in each category.

**Table 6 sensors-23-09873-t006:** The results of the statistical analysis (mean and standard deviation (SD)) of all four approaches (B-mode, SEGL, Gabor, and LBP) between all 3 groups: A-MTrPs, L-MTrPs, and healthy controls.

	Approach	*p*-Value	Mean (A-MTrPs)	SD (A-MTrPs)	Mean (Healthy)	SD (Healthy)	Mean (L-MTrPs)	SD (A-MTrPs)
Entropy	Gabor	$2.32\;\times \;10^{-2}$	$7.30\;\times \;10^{-4}$	$1.58\;\times \;10^{-4}$	$6.23\;\times \;10^{-4}$	$2.00\;\times \;10^{-4}$	$7.40\;\times \;10^{-4}$	$1.77\;\times \;10^{-4}$
	SEGL	$1.70\;\times \;10^{-2}$	$7.67\;\times \;10^{-2}$	$2.96\;\times \;10^{-2}$	$5.34\;\times \;10^{-2}$	$3.04\;\times \;10^{-2}$	$7.19\;\times \;10^{-2}$	$3.72\;\times \;10^{-2}$
	B-mode	$6.88\;\times \;10^{-1}$	6.19	4.17 ×−10	5.72	4.12 ×−10	6.10	$4.58\;\times \;10^{-1}$
	LBP	$1.00\;\times \;10^{-3}$	5.37	1.75 ×−10	5.36	2.47 ×−10	5.36	$1.81\;\times \;10^{-1}$
Energy	Gabor	$1.00\;\times \;10^{-3}$	$9.36\;\times \;10^{8}$	$1.83\;\times \;10^{8}$	$1.13\;\times \;10^{9}$	$2.87\;\times \;10^{8}$	$9.31\;\times \;10^{8}$	$2.05\;\times \;10^{8}$
	SEGL	$1.38\;\times \;10^{-2}$	$-2.27\;\times \;10^{8}$	$9.94\;\times \;10^{7}$	$-1.86\;\times \;10^{8}$	$4.25\;\times \;10^{7}$	$-2.17\;\times \;10^{8}$	$6.56\;\times \;10^{7}$
	B-mode	$6.38\;\times \;10^{-2}$	$1.73\;\times \;10^{8}$	$9.50\;\times \;10^{7}$	$9.28\;\times \;10^{7}$	$5.42\;\times \;10^{7}$	$1.47\;\times \;10^{8}$	$9.62\;\times \;10^{7}$
	LBP	$1.40\;\times \;10^{-3}$	$1.40\;\times \;10^{9}$	$2.57\;\times \;10^{8}$	$1.56\;\times \;10^{9}$	$4.37\;\times \;10^{8}$	$1.37\;\times \;10^{8}$	$2.68\;\times \;10^{8}$
Mean	Gabor	$2.34\;\times \;10^{-1}$	$1.23\;\times \;10^{2}$	1.58	$1.24\;\times \;10^{2}$	1.26	$1.23\;\times \;10^{2}$	$1.64\;\times \;10^{4}$
	SEGL	$1.38\;\times \;10^{-2}$	$9.74\;\times \;10^{-3}$	$4.13\;\times \;10^{-3}$	$6.47\;\times \;10^{-3}$	$4.10\;\times \;10^{-7}$	$9.17\;\times \;10^{-3}$	$5.20\;\times \;10^{-3}$
	B-mode	$4.20\;\times \;10^{-3}$	4.72 × 10	1.73 × 10	3.08E × 10	1.03 × 10	4.31 × 10	1.80 × 10
	LBP	$1.40\;\times \;10^{-3}$	$1.17\;\times \;10^{2}$	7.90	$1.09\;\times \;10^{2}$	1.00 × 10	$1.16\;\times \;10^{2}$	9.89
Contrast	Gabor	$3.80\;\times \;10^{-3}$	$4.13\;\times \;10^{11}$	$5.36\;\times \;10^{10}$	$4.62\;\times \;10^{11}$	$7.85\;\times \;10^{10}$	$4.11\;\times \;10^{11}$	$5.95\;\times \;10^{10}$
	SEGL	$2.04\;\times \;10^{-2}$	$7.65\;\times \;10^{6}$	$3.53\;\times \;10^{6}$	$4.95\;\times \;10^{6}$	$3.57\;\times \;10^{6}$	$7.09\;\times \;10^{6}$	$4.42\;\times \;10^{6}$
	B-mode	$1.90\;\times \;10^{-1}$	$1.59\;\times \;10^{11}$	$5.37\;\times \;10^{10}$	$1.23\;\times \;10^{11}$	$4.22\;\times \;10^{10}$	$1.41\;\times \;10^{11}$	$5.05\;\times \;10^{10}$
	LBP	$2.01\;\times \;10^{-2}$	$3.98\;\times \;10^{11}$	$4.96\;\times \;10^{10}$	$4.20\;\times \;10^{11}$	$7.70\;\times \;10^{10}$	$3.93\;\times \;10^{11}$	$4.90\;\times \;10^{10}$
Homogeneity	Gabor	$1.7\;\times \;10^{-3}$	$4.58\;\times \;10^{4}$	$9.30\;\times \;10^{3}$	$5.54\;\times \;10^{4}$	$1.44\;\times \;10^{4}$	$4.57\;\times \;10^{4}$	$1.05\;\times \;10^{4}$
	SEGL	$1.02\;\times \;10^{-2}$	39.76	1.18 × 10	2.9 × 10	1.20 × 10	3.8 × 10	1.52 × 10
	B-mode	$3.24\;\times \;10^{-2}$	$1.74\;\times \;10^{4}$	$6.61\;\times \;10^{3}$	$1.30\;\times \;10^{4}$	$5.60\;\times \;10^{3}$	$1.54\;\times \;10^{4}$	$6.69\;\times \;10^{3}$
	LBP	$3.14\;\times \;10^{-2}$	$4.30\;\times \;10^{4}$	$8.12\;\times \;10^{3}$	$4.82\;\times \;10^{4}$	$1.40\;\times \;10^{4}$	$4.11\;\times \;10^{4}$	$8.63\;\times \;10^{3}$
Correlation	Gabor	$8.56\;\times \;10^{-2}$	$-2.27\;\times \;10^{8}$	$9.94\;\times \;10^{7}$	$-1.86\;\times \;10^{8}$	$4.26\;\times \;10^{7}$	$-2.17\;\times \;10^{8}$	$6.56\;\times \;10^{7}$
	SEGL	$3.10\;\times \;10^{-2}$	$1.39\;\times \;10^{9}$	$1.89\;\times \;10^{8}$	$1.51\;\times \;10^{9}$	$2.56\;\times \;10^{8}$	$1.39\;\times \;10^{9}$	$1.43\;\times \;10^{8}$
	B-mode	$3.00\;\times \;10^{-4}$	$1.05\;\times \;10^{7}$	$2.71\;\times \;10^{7}$	$7.01\;\times \;10^{7}$	$5.83\;\times \;10^{7}$	$1.90\;\times \;10^{7}$	$3.14\;\times \;10^{7}$
	LBP	$2.00\;\times \;10^{-5}$	$-5.08\;\times \;10^{6}$	$7.12\;\times \;10^{5}$	$-3.88\;\times \;10^{6}$	$1.33\;\times \;10^{6}$	$-4.86\;\times \;10^{6}$	$9.44\;\times \;10^{5}$
Variance	Gabor	$6.00\;\times \;10^{-4}$	$2.71\;\times \;10^{7}$	$5.72\;\times \;10^{6}$	$3.18\;\times \;10^{7}$	$6.45\;\times \;10^{6}$	$2.63\;\times \;10^{7}$	$4.96\;\times \;10^{6}$
	SEGL	$1.40\;\times \;10^{-2}$	$6.30\;\times \;10^{2}$	$2.66\;\times \;10^{2}$	$4.19\;\times \;10^{2}$	$2.64\;\times \;10^{2}$	$5.93\;\times \;10^{2}$	$3.35\;\times \;10^{2}$
	B-mode	$4.70\;\times \;10^{-3}$	$3.06\;\times \;10^{7}$	$1.40\;\times \;10^{7}$	$1.97\;\times \;10^{7}$	$1.01\;\times \;10^{7}$	$2.80\;\times \;10^{7}$	$1.13\;\times \;10^{7}$
	LBP	$1.70\;\times \;10^{-3}$	$5.94\;\times \;10^{8}$	$1.12\;\times \;10^{8}$	$7.04\;\times \;10^{8}$	$1.83\;\times \;10^{8}$	$5.93\;\times \;10^{8}$	$1.36\;\times \;10^{8}$

Note: The bolded numbers represent the statistical significance (*p* < 0.05).

## Data Availability

The data collected and analyzed in this study are available from the corresponding authors upon reasonable request.

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
