# Peer review of "Identification of Myofascial Trigger Point Using the Combination of Texture Analysis in B-Mode Ultrasound with Machine Learning Classifiers"

_sensors, 2023, doi:10.3390/s23249873_

Round 1

Reviewer 1 Report

Comments and Suggestions for Authors

Using the Combination of Texture Analysis in B-mode Ultrasound, the authors compared different ML classifiers to identify Myofascial Trigger Points. Following are the comments that need to be addressed in the revised version of the paper.

1.       Line 19: what is the meaning of the three methods calculated for each image? Does it mean three types of features are extracted? SEGL must be defined first before using it.

2.       Line 20: What is the meaning of "Then, the statistical features (e.g., entropy) were calculated over the B-mode-US images and those three methods."

3.       The abstract needs re-writing to describe clearly what has been done.

4.       Line 47: what is the US? Always describe the full names before using the abbreviations.

5.       Line 62: Earlier, the authors explained using Doppler and elastography ultrasound images to distinguish the affected muscle. Isn't it a gold standard for the classification?

6.       Line 67: It is better if all these four categories of techniques may be compared, and the authors must highlight the pros and cons of these techniques. Summarizing in the form of tables will be very useful. Moreover, a comparison of performance on different datasets should be done in the literature review.

7.       Table 2: Why equations are shown in terms of a table. They can be explained in the text with the help of a figure. Several symbols in the equations are not defined.

8.       In Figure 2, what is the purpose of showing the Gabor feature for an MTrP subject and LBP for a healthy subject? It is better to show and identify the differences in the Gobor, LBP, and other feature maps.

9.       Figure 3: What are the RED color connections? How (40x8) is 280?

10.   Table 3: Explain and cite all the ML classifiers in this table. How are these classifiers selected from many ML classifiers in the literature?

11.   Line 266: Explain (256x512x128) in the NN. Is 256 the size of the input layer? What are these 256 inputs?

12.   Line 272: Describe the inputs to the ML classifiers more clearly.

13.   Line 274: How the training and testing datasets are divided is unclear. It will be helpful if the authors make a table to show how many data points are used for training, validation, and testing.

14.   Line 278: The ROC curve and area under the ROC curve can also be good performance metrics.

15.   Line 281: Two "And" are in the title. The ensemble has a meaning in ML, a combination of classifiers not reflected in this section.

16.   Line 188: What is the meaning of leave on feature out approach? Typically, different combinations of features are used without mentioning leaving one out.

17.   Section 3: This section is poorly written. Results are reported in Tables 4 and 5, but the explanation is unclear. A better and more detailed description is required. How is the majority voting used? The majority voting is not explained in the methodology section. Training, validation, and testing are not described in the results section. Results should be reported as mean and standard deviation in Table 4.

18.   What is the effect of parameters in ML classifiers, and how the parameters are optimized?

19.   It is essential to compare the reported results with the published literature in this domain. I think there are many papers in this area. So, it is necessary to describe how the reported results are better than the published results and the novelty of the work. Two of the papers are mentioned below,

Ryan G.L. Koh, Banu Dilek, Gongkai Ye, Alper Selver, Dinesh Kumbhare, Myofascial Trigger Point Identification in B-Mode Ultrasound: Texture Analysis Versus a Convolutional Neural Network Approach, Ultrasound in Medicine & Biology, Volume 49, Issue 10, 2023,Pages 2273-2282

Behr M, Saiel S, Evans V, Kumbhare D. Machine Learning Diagnostic Modeling for Classifying Fibromyalgia Using B-mode Ultrasound Images. Ultrasonic Imaging. 2020;42(3):135-147. doi:10.1177/0161734620908789

Comments on the Quality of English Language

Extensive proofreading is required.

Author Response

Thanks!

Reviewer 2 Report

Comments and Suggestions for Authors

Dear Authors,

I congratulate you on your very complex study about identifying muscular trigger points on ultrasound images using machine learning.

However there are some aspects that require your attention.

At line 5 write the complete names of all authors.

You need to better underline the possible future use of these findings in various medical fields of activity: orthopedics, kineto-therapy, imaging.

Also insert a paragraph of future development at different levels of the muscular system.

At the end of the manuscript insert a list of abbreviations.

Looking forward to see your manuscript published.

Author Response

Dear Reviewer, 

Thanks!

Round 2

Reviewer 1 Report

Comments and Suggestions for Authors

The authors have addressed all of my comments in the revised paper.